# A Fault Detection System for a Geothermal Heat Exchanger Sensor Based on Intelligent Techniques

**DOI:** 10.3390/s19122740

**Published:** 2019-06-18

**Authors:** Héctor Aláiz-Moretón, Manuel Castejón-Limas, José-Luis Casteleiro-Roca, Esteban Jove, Laura Fernández Robles, José Luis Calvo-Rolle

**Affiliations:** 1Departamento de Ingeniería de Sistemas y Automática, Universidad de León, 24071 León, Spain; 2Departamento de Ingenierías Mecánica, Informática y Aeroespacial, Universidad de León, 24071 León, Spain; manuel.castejon@unileon.es (M.C.-L.); l.fernandez@unileon.es (L.F.R.); 3Departamento de Ingeniería Industrial, Universidade da Coruña, 15405 Ferrol, Spain; jose.luis.casteleiro@udc.es (J.-L.C.-R.); esteban.jove@udc.es (E.J.); jlcalvo@udc.es (J.L.C.-R.)

**Keywords:** fault detection, geothermal heat exchanger, random decision forests, gradient boosting, extremely randomized trees, adaptive boosting, k-nearest neighbors, shallow neural networks

## Abstract

This paper proposes a methodology for dealing with an issue of crucial practical importance in real engineering systems such as fault detection and recovery of a sensor. The main goal is to define a strategy to identify a malfunctioning sensor and to establish the correct measurement value in those cases. As study case, we use the data collected from a geothermal heat exchanger installed as part of the heat pump installation in a bioclimatic house. The sensor behaviour is modeled by using six different machine learning techniques: Random decision forests, gradient boosting, extremely randomized trees, adaptive boosting, k-nearest neighbors, and shallow neural networks. The achieved results suggest that this methodology is a very satisfactory solution for this kind of systems.

## 1. Introduction

In recent years, most countries faced an important challenge in terms of global warming, economic instability and fossil fuels price dependency. In this context, the use of alternative energies has been promoted by the administrations. The most common alternative energy sources are the wind and solar energies, whose technologies have been subjected to significant advances. However, in addition to these two energies, the promotion of other renewable energies, such as oceanic or geothermal energy, have presented important developments in terms of efficiency [1].

Geothermal energy is defined as the heat energy stored under the ground. Dickson and Fanelli, in [2], presented an estimation of the amount of heat inside the earth rounds the $42\times 10^{12}$ W. In spite of this high amount of energy, geothermal installations must be placed in specific areas with suitable geological conditions [3]. Around the world, its use represents 15 MW of non electrical applications, such as industrial processes, bathing or heat pumps, and 9 MW of the electrical ones.

The heat exchanger is a crucial component of a geothermal facility, and its main function is to absorb heat from the ground or transfer it. A geothermal heat exchanger can be placed under the ground in vertical or horizontal configurations [4,5]. On the one hand, vertical configurations are more efficient because at high depths, the ground temperature remains almost constant along the year. This means that, compared to the ambient temperature, the ground temperature would be higher in winter and lower in summer. On the other hand, horizontal configurations are less expensive, since the setup is simpler.

In this kind of facility, where the energy efficiency plays a significant role, the appearance of any kind of anomaly may lead to inefficient performance. Hence, in renewable energy systems, or any industrial plant in general terms, the anomaly detection is a crucial task [6,7,8,9,10,11]. These anomalies can be produced by wrong sensor readings, actuator malfunctions or changes in plant parameters, in general terms [12,13,14,15,16,17,18]. Focusing on the sensors performance, the occasional reading errors can be removed and recovered, making the systems more fault tolerant and robust [19,20,21,22,23,24,25].

This work deals with a geothermal heat pump facility used to provide thermal energy from the ground [26]. To achieve a geothermal system optimization, the good behaviour of the system equipment must be ensured. Then, the prediction of the correct sensor values is a key step to perform a proper fault identification and recovery. An anomaly would lead to a high deviation from the real and predicted value. In this case, the real measurement would be discarded and the value considered by the control system would be the predicted one.

With the aim of improving system performance, machine learning techniques are commonly considered. These techniques rely on actual observations registered from the system that are used to train the model. In this work, the sensor reading prediction is performed using intelligent models, trained with data from a geothermal heat pump installation. Four different intelligent techniques, commonly used for these kind of applications, were applied to the dataset: Shallow neural networks, extremely randomized trees, random decision forest and gradient boosting. Two possible approaches can be considered to obtain the best model. The first approach uses the whole dataset to train a global model. The second approach is based on the division of the dataset to apply the intelligent regression techniques over each group. In all cases, the models were tested using artificially generated outliers, obtaining successful results.

The document is organized following this structure: The next section describes the case of study. Then, Section 3 details the proposed fault detection and recovery system. The experiments performed and the obtained results are presented in Section 4. The results are discussed in Section 5 and, finally, the conclusions are explained in Section 6.

## 2. Case of Study

This section describes the geothermal heat exchanger facility under study located in a bioclimatic house.

### 2.1. Sotavento Bioclimatic House

The Sotavento bioclimatic house is a building dedicated to promote the use of alternative energies and the energy savings. These facilities, founded by the Sotavento Galicia Foundation, are located between the councils of Xermade and Monfero (Lugo), in the autonomous community of Galicia (Spain). Its geographical coordinates are 43${}^{\circ }$21${}^{\prime }$ North, 7${}^{\circ }$52${}^{\prime }$ West, with an elevation of 640 m above the sea and at a distance of 30 km from the sea.

Two different energy needs must be satisfied in the Sotavento bioclimatic house: The thermal and the electric energy. The thermal system has three different renewable energy sources: Geothermal, solar and biomass. These three sources ensure the thermal demand coverage. The thermal installation can be divided into three parts [27]:Generation group: Three different renewable sources are exploited:
-Geothermal system: A horizontal collector consisting of 5 loops of 100 m is placed under the ground at a depth of 2 m. The heat pump is a MAMY Genius—10.3 kW, and it has a nominal electrical power consumption of 1.9 kW and a nominal thermal power of 8.4 kW. The energy is absorbed from the ground and it is used to heat a mixture of water and glycol.-Solar thermal: Eight solar panels absorb the solar radiation to heat the ethyleneglycol flowing inside them.-Biomass boiler system: A biomass boiler type Ökofen, model Pallematic 20, with a configurable power of 20 kW, with a yield of pellets of 90%.Energy accumulation group: The thermal energy storage is ensured using different accumulators. A solar accumulator of 1000 L receives the thermal energy from the solar system. In series, an inertial accumulator of 800 L stores the heat from the boiler and geothermal systems.Consumption group: The thermal system must cover the demand of underfloor heating systems and Domestic Hot Water (DHW). The underfloor heating system is designed to keep the house temperature between 18 ${}^{\circ }$C and 22 ${}^{\circ }$C. The fluid temperature should remain between 35 ${}^{\circ }$C and 40 ${}^{\circ }$C. According to the Spanish Technical Building Code, the DHW demands 240 L/day.

In addition to the thermal systems, the Sotavento bioclimatic house has also an electrical installation with two renewable sources: Wind and photovoltaic. The electricity supplies the power systems and the lighting. To avoid power cuts, the house is connected to the power grid when it is demanded.

### 2.2. The Geothermal System

A more detailed explanation about the geothermal energy system is presented in this subsection. It is divided into two main parts described below: The heat pump and the heat exchanger (Figure 1).

*Heat Pump.* The Heat Pump has two different circuits; the primary one provides the heat from the ground (the geothermal exchanger) to the heat pump unit, and the other one is connected between the unit and the inertial accumulator. The energy absorbed from the ground is measured by two sensors.

*Geothermal exchanger.* The horizontal exchanger consists of five different circuits. The ground temperature along the exchanger is monitored using sensors distributed in four different loops. A scheme of the sensors located along the exchanger can be seen in Figure 2. Sensors S28 and S29 measure the energy absorbed from the ground and S401 measures the ground temperature. The rest of the sensors monitor the exchanger temperature in different points.

### 2.3. The Dataset

The initial dataset corresponds to the temperatures measured by the sensors during one year, registered with a sample time of 10 min.

Sensors S28 and S29 (the input and the output of the heat exchanger) are located inside the house. Hence, when the heat pump is off, these sensors measure the temperature inside the house. For this reason, the temperature is filtered to take into account only the data when the heat pump is on.

As this work proposes a model capable of predicting the sensor measurements to detect anomalies, only data from correct operation is considered. Then, to avoid wrong samples (bad sample time, bad range, open wires, etc.), the dataset was filtered to discard the erroneous data. After this conditioning step, the samples were reduced from 52,705 to 52,699.

However, as the appearance of any kind of anomaly in a sensor must be detected in a short time, the models were implemented with an amount of data corresponding to two days. These measurements are randomly selected from the 52,699 samples.

## 3. Fault Detection and Recovery (FDR) Approach—Used Techniques

The scheme defined for fault detection and recovery approach is shown in Figure 3. It is possible to divide the figure into two parts: The model and the fault detection and recovery block. The first one gives the prediction of each sensor based on the measurements made by the rest of the sensors. The second one compares the prediction with the real measurement, and analyzes the deviation based on a defined range. If there is a significant deviation, the valid signal is the prediction. Otherwise, the real measurement is set at the output.

### 3.1. FDR Steps

In this subsection, the necessary steps to accomplish the FDR developed approach are explained.

*Sensor fault detection.* Initially, a simple methodology for accomplishing sensor fault detection technique is used. The method allows a specific configuration of the range deviation. If the measured sample is out of this range, then a fault is labeled. The deviation percentage is referred to the operating temperature range.

*Recovery.* If a fault is detected, then it is necessary to recover the wrong sample with a value prediction. This prediction could be based on the other sensors readings, their previous values, and so on. To accomplish the recovery, a model must be implemented with the aim to predict an accurate value.

### 3.2. Used Techniques

The present subsection shows the different techniques used for accomplishing the objectives of the present research.

#### 3.2.1. Analysis and Preprocessing

From the considered initial data, two different subclasses were created:Day data cases.Night data cases.

Knowing each date of the data recollection and the precise location of the installation under study, the sunrise and sunset times can be obtained. This is the criteria used to split the data in the two subclasses.

To obtain a representative model, some variables of the raw dataset have been selected. In addition, the previous state of some signals is included as an artificial input, for developing each experiment shown in Section 4.

The use of this extra information can be more beneficial than obtaining the model with original data features only. The election of these artificial features is always based on expert knowledge about the system behavior [28].

Based on a data description of the new dataset generated from the raw data, a common pre-processing procedure has been developed, including those experiments with previous values of different sensor like artificial variables.

The criterion for data normalizing is shown in Equation (Equation 1):
(1)$$\frac{X_{i}-mean\left(x\right)}{stdev(x)}$$

The Standard Scaler data input pre-processing has been implemented with Python *sklearn.preprocessing. StandardScaler* [29] library. The main goal of the normalization step is to avoid the very soon convergence in the first iterations, when the training process of a particular regression method begins [30].

#### 3.2.2. Regression Techniques

The recovery methodology purpose is to mimic the actual behaviour of the sensor. Thus, a predictive model trained with data acquired from the sensor is a sensible approach for achieving a computational representation of the sensor. Six different types of predictive models have been tested: Shallow neural networks, extremely randomized trees (ExtraTrees), random decision forests, adaptive boosting, k-nearest neighbors, and gradient boosting.

This choice of regressors pursues to represent the complexity of the sensor’s behaviour by two subtly different approaches: The shallow neural network solution features a single model capable of increasing its complexity by means of enlarging the number of neurons in its hidden layer; on the other hand, the extremely randomized trees, adaptive boosting, random decision forests, and gradient boosting regressors, belong to the ensemble methods family. Ensemble models provide their results by combining those obtained from multiple elementary models. In this case, complexity is approached by enlarging the number of simple models comprised in the ensemble.

The Multilayer Perceptron (MLP) is one of the most frequently used shallow neural network architectures. The good performance of this kind of artificial neural network has been proven in similar works such as [31,32,33]. Previous research [34] proved how this technique is capable of providing satisfactory results in the context of much larger amounts of data than those used in the case of study.

Ensemble methods, on the other hand, are among the most frequently used techniques for the excellent results they usually display. Examples of successful stories can be discovered following Kaggle’s machine learning competitions (https://www.kaggle.com/), where, along with Deep Neural Networks, ensemble methods such as those reported in this research are most frequently the winning techniques.

Each technique and the set of their associated parameters used in this work are explained bellow:*Shallow Neural Networks.* Artificial Neural Networks can be used as universal approximators [35]. For this paper, a three layer Multi Layer Perceptron architecture was chosen: An input layer for capturing the sensor information, a hidden layer with non linear activation functions, and an output layer with one single neuron and a linear activation function to provide the prediction. The most important hyperparameters governing the regressor performance are the hidden layer size, the maximum number of iterations, the early stopping, the activation function, the nesterov momentum and the solver.*K-Nearest Neighbors.* This is a representative of instance based techniques or non generalizing learning. Instead of representing the data via a model, this technique stores instance and uses a voting scheme on the nearest neighbors for obtaining the prediction on new data. This technique is a popular choice for setting a baseline for the prediction error. The most important hyperparameter is the number of neighbors.*Adaptive Boosting.* This technique belongs to the stagewise additive models family. The prediction is based on a weighted sum of the simpler weak estimators it comprises. Each weak estimator is designed to concentrate on those samples that previous estimators found still to be difficult to fit. In this technique, the number of estimators is the most important hyperparameter to tune.*Random Decision Forests.* Being one of the most popular ensemble methods, Random decision forests (RF) comprise a collection of simple decision trees whose results are considered to emit a final collective result. RF basic components can be built by considering a random limited number of features and/or a random limited number of observations. Thus, each component only has access to a fraction of the information and pays attention to specific details in the portion of information assigned to them. The combination of a number of these simple basic trees most frequently outperforms the results from a larger and more complex single tree. The number of estimators is the most important hyperparameter to tune.*Extremely Randomized Trees.* They are similar to Random Forests, as they combine an ensemble of decision trees. Nevertheless, a few important differences are worth noting: Firstly, Extra Trees can provide piece-wise multilinear approximations to the training dataset instead of the piece-wise constants one provides by random forests. Secondly, Extra Trees are based on using random values for the optimal cut point choice, instead of bootstrapping to find the optimal cut point [36]. Similarly to RF, one of the most important hyperparameters to tune is the number of basic estimators.*Gradient Boosting.* This technique builds the model following a stage-wise approach, by adding subsequent basic estimators in order to capture the unexplained information present in the residuals of former weak estimators [37]. The estimators frequently are decision trees and, similarly, the number of basic estimators is among the most important hyperparameters.

## 4. Experiments and Results

This section describes the different experiments carried out and the results obtained.

### 4.1. Experiment Definition

Depending on the predictors used in the predictive model, four different experiments are defined:Experiment A: Prediction of sensor S-315 based on S-309 to S-316 signalsExperiment B: Prediction of sensor S-315 based on S-309 to S-316 signals and their previous statesExperiment C: Prediction of sensor S-315 based on S-309 to S-316 signals and S-315 previous stateExperiment D: Prediction of sensor S-315 based on S-309 to S-316 signals, their previous states, and S-315 previous state

In each experiment, the four regression techniques mentioned above —shallow neural networks, extremely randomized trees, random decision forests, and gradient boosting— are used to build two types of models, according to the data used for each one:
Global models: In this case the whole data set is used for training a single regressor.Hybrid models: In this case, the data set is split into two groups in accordance to day and night criteria. Two different models are fit, one for day usage and another one for the night hours.

### 4.2. Error Metrics

In order to compare the different regression models obtained, the following error metrics have been implemented:MAE: Mean Absolute Error. The goal of this metric is to measure the difference between predicted and real values. This metric has some advantages compared to other error measures [38].
(2)$$MAE=\frac{1}{m}\sum_{k=1}^{m}\left|Y_{k}-{\hat{Y}}_{k}\right|$$ where $Y_{k}$ is the observed value and ${\hat{Y}}_{k}$ is the foretold value.LMLS: Least Mean Log Squares. This metric is used as regression loss function in the training process as well as in the validation error measure [39], Equation (Equation 3).
(3)$$L.M.L.S=\frac{1}{m}\sum_{k=1}^{m}\log\left(1+\frac{1}{2}{\left(Y_{k}-{\hat{Y}}_{k}\right)}^{2}\right)$$ where $Y_{k}$ is the observed value and ${\hat{Y}}_{k}$ is the foretold value.SMAPE: Symmetric Mean Absolute Percentage Error. The main goal of this metric is to explain relative errors thanks to the use of percentages [40], Equation (Equation 4).
(4)$$S.M.A.P.E=\frac{2}{m}\sum_{k=1}^{m}\frac{|Y_{k}-{\hat{Y}}_{k}|}{Y_{k}+{\hat{Y}}_{k}}$$ where $Y_{k}$ is the observed value and ${\hat{Y}}_{k}$ is the foretold value.MSE: Mean Squared Error. This metric can include the variance of error, it can be applied in several forecasting problems [41] Equation (Equation 5).
(5)$$M.S.E.=\frac{1}{m}\sum_{k=1}^{m}{\left(Y_{k}-{\hat{Y}}_{k}\right)}^{2}$$ where $Y_{k}$ is the observed value and ${\hat{Y}}_{k}$ is the foretold value.MAPE: Mean Absolute Percentage Error. This error metric is one of the most common measures of the accuracy in regression problems [42], Equation (Equation 6).
(6)$$M.A.P.E=\frac{100\%}{m}\sum_{k=1}^{m}\frac{|Y_{k}-{\hat{Y}}_{k}|}{Y_{k}}$$ where $Y_{k}$ is the observed value and ${\hat{Y}}_{k}$ is the foretold value.NMSE: Normalised Mean Square Error. This a measure oriented to estimate the overall deviations between observed and predicted values [43], Equation (Equation 7).
(7)$$N.M.S.E=\frac{1}{m}\sum_{k=1}^{m}\frac{{\left({\hat{Y}}_{k}-Y_{k}\right)}^{2}}{mean\left({\hat{Y}}_{k}\right)*mean\left(Y_{k}\right)}$$ where $Y_{k}$ is the observed value and ${\hat{Y}}_{k}$ is the foretold value.

### 4.3. Experiments Setup

For each experiment the dataset was split into two subsets—training and test sets—as customary in data science projects in order to provide the error value on a held out dataset. Such an error represents the capability of the method to generalize the observed behavior to new unseen data. Thus, a fraction comprising 70% of the samples is used for training purposes to adjust the parameters of the models, while a fraction with 30% of the samples is used for final testing. In order to find the best combination of hyperparameters for each model, a grid search with ten fold cross validation has been carried out. The chosen scoring criteria was the negative mean square error. As a preprocessing step, the data is normalized before entering the regression model. In order to avoid leaking information from the validation test during cross validation, both the scaler and the regressor are embedded in a pipeline.

The four families of regressors, the scaler, the pipeline tool, and the grid search with cross validation, are implemented in Scikit-Learn’s machine learning library [44] which provides easy access to these techniques using Python as programming language for computational purposes.

The search space for the best values of the hyperparameters is reported below. Those hyperparameters not mentioned adopt Scikit-Learn default values.

#### 4.3.1. Shallow Neural Network


hidden_layer_sizes=[(n,) for n in ( 5, 6, 7, 8)]

max_iter=[ 500_000]

learning_rate_init=[1e-1, 1e-2, 1e-3]

early_stopping=[True]

activation=[’relu’]

nesterovs_momentum=[True]

warm_start=[False]

solver=[’lbfgs’]


#### 4.3.2. Extremely Randomized Tree


n_estimators=range(10, 100, 5)


#### 4.3.3. Random Decision Forests


n_estimators=range(10, 100, 5)


#### 4.3.4. Gradient Boosting


n_estimators=range(10, 100, 5)

learning_rate=np.linspace(1e-3, 1e-1, 5)

n_iter_no_change=[2]


#### 4.3.5. AdaBoost


n_estimators=range(10, 100, 5)


#### 4.3.6. K-Neareat Neighbors


n_neighbors=range(5, 20, 5)


Table 1, Table 2 and Table 3 show the results obtained in the experiments by the global and hybrid approaches (best ones in bold). According to most error metrics, the ExtraTrees regressor achieves the best results in both global and hybrid approaches. Among these two, the hybrid approach displays better results, particularly according to the mean absolute error criteria, the easiest to interpret by human beings. Figure 4 and Figure 5 display the results obtained by the six types of regressors considered, in this case using the data from experiment A. It is clear that the Extra Trees regressor achieves great resemblance with the actual data recorded from the sensor in what are considered very satisfactory results.

## 5. Discussion

The number of experiments, regressors and error metrics reported in this paper builds a complex scenario when attempting to establish a single winner solution. As it usually happens in real engineering problems, the solution to a problem is not unique and the context determines the preferred one.

From a strictly numerical point of view, it could be argued that Experiment A frequently displays best error values. In those cases where it fails to outperform other experiments, the results are not significantly different from the minimum.

Considering the experiments from the point of view of their complexity, Experiment A also represents the simplest configuration as it requires the lowest number of input variables; a fact that, in absence of significant differences in performance with respect to the rest of the experiments, also advocates for its designation as the preferred configuration.

In economic terms, the context around this study case does not justify a more complex configuration. In some scenarios, e.g., optimizing a quality feature in manufacturing processes, a marginal improvement in the prediction model leads to significant economic benefits; but that is hardly the case of the study case reported in this paper: A predictive model that is used as a backup for the real sensor and whose reads are only considered during malfunctioning.

According to these former criteria, Experiment A could be considered the best choice, but considering the following practicalities, the final decision might differ. Firstly, an important issue to consider is the intrinsic precision of the actual sensor being modeled. If the performance difference between alternatives is orders of magnitude lower than the sensor precision, then those alternatives are in fact equally optimal. Secondly, the results must be considered from the point of view of the subsequent data consumption. If the sensor data is to be further processed by an algorithm sensitive to a specific precision, it makes little sense considering differences in considerable smaller differences, e.g., comparing two temperature readings with values 82.15 F and 82.17 F when the on–off controller driving a pump already made a decision at a 60 F threshold. This paper deliberately does not specify the particular subsequent model that the sensor signal feeds, as many such systems can be considered. Essentially, it is the engineer’s call to weigh the context factors and choose the optimal solution for the problem at hand, the numerical error scored by each alternative being an important but not unique criterion in the decision making process. For the study case reported in this paper, Experiment A using Extra Trees was adopted as the preferred solution. Nevertheless, the approach proposed relies on training data from a short period of time, two days, which makes it possible to periodically retrain the models and perform the comparison to select the new best choice and adapt for future changes.

## 6. Conclusions and Future Works

A methodology for recovering data missing in malfunctioning state sensor and the sensor fault detection have been addressed in this research successfully. Sensor fault detection procedure is relaying on tagging data as fault, when a measured sample is out of the range derivation. Moreover, the procedure for recovering data missing is based on the implementation of several experiments with the aim to get the best way to define a model when it is trying to get measurements of a sensor with problems. Input data features election is relevant when a robust regression model wants to be created to predict missing data in a process where the temperature is involved—more concretely, the election of new features and how these are estimated or calculated. In this research, new artificial features based on the sensor values on the previous state are added to achieve and compare a global model and hybrid model for recovering missing data of a sensor. Results prove that a hybrid model implemented with an Extremely Ramdomized Trees regressor, composed by day and night submodels not including previous state values as artificial features, is the best way for recovering data missing. Future works will explore the improvement of the sensor fault detection procedure via anomaly detection techniques such as Isolation Forest, One Class SVM (Support Vector Machines), Local Outlier Factor, and Elliptic Envelope. From the point of view of recovering missing data, new experiments based on time series oriented to prevent the use of previous state information will be implemented. Some new, complex and data fusion models will be used also in the next research phase.

## Figures and Tables

**Figure 1 sensors-19-02740-f001:**
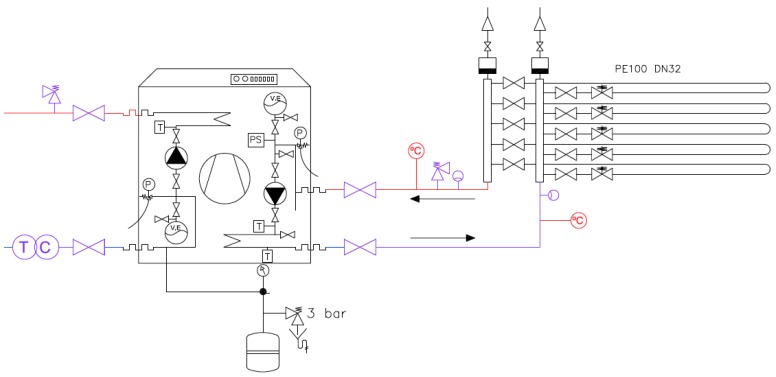
Heat pump and horizontal exchanger layout.

**Figure 2 sensors-19-02740-f002:**
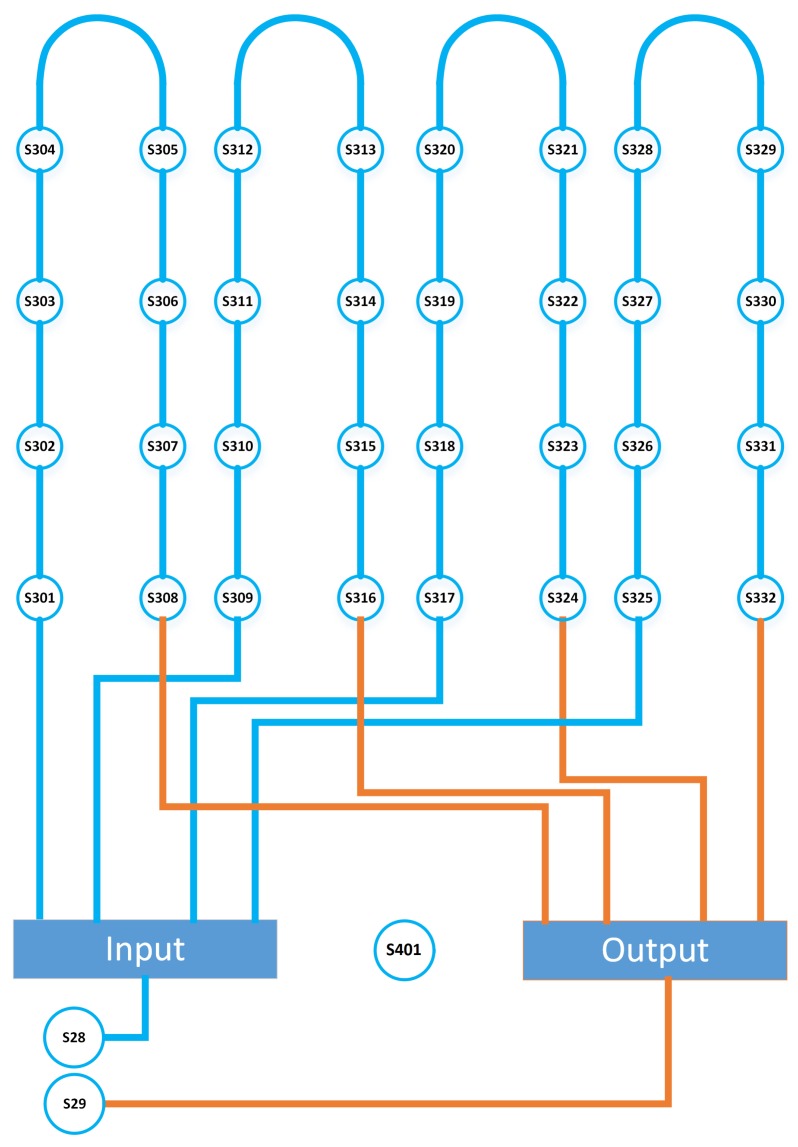
Geothermal exchanger sensors layout.

**Figure 3 sensors-19-02740-f003:**
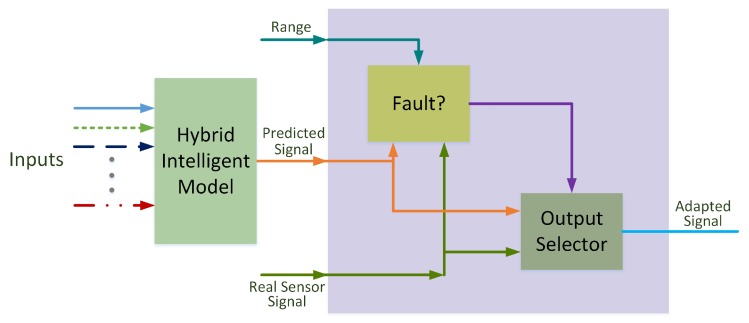
Fault detection and recovery approach.

**Figure 4 sensors-19-02740-f004:**
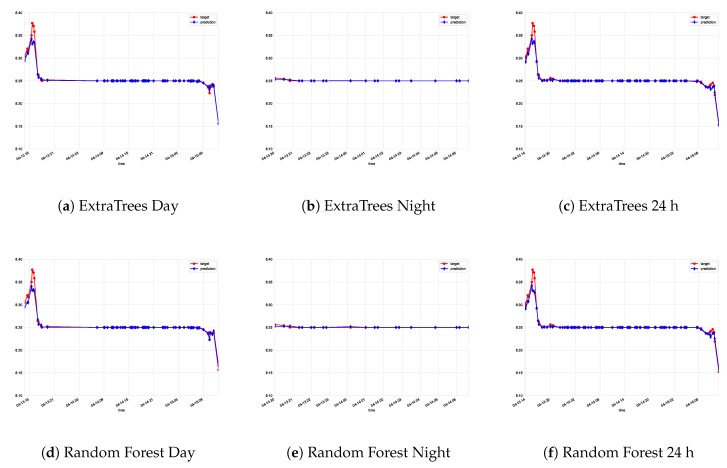
Actual data vs. predictions for Experiment A. ExtraTrees, random forest, gradient boosting, and MLP are considered for each model (day, night, and global model).

**Figure 5 sensors-19-02740-f005:**
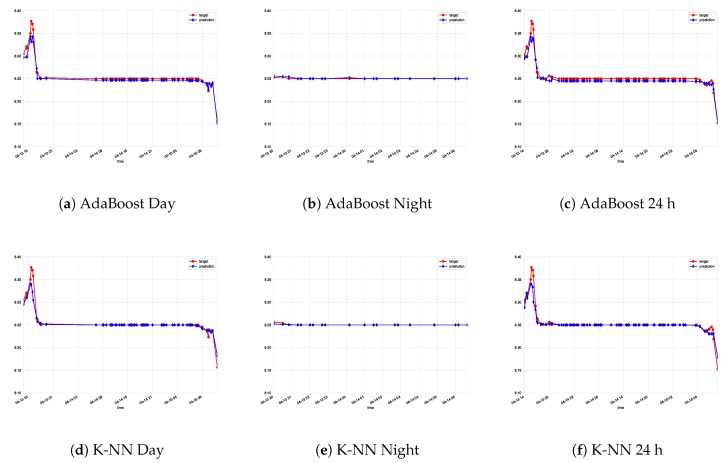
Actual data vs. predictions for Experiment A. AdaBoost and k-nearest neighbors are considered for each model (day, night, and global model).

**Table 1 sensors-19-02740-t001:** Global model errors (multiplied by $10^{5}$) for extremely randomized trees (ET), gradient boosting (GB), multi-layer perceptron (MLP), random forest (RF), adaptive boosting (AB), and k-nearest neighbors (K-NN).

Error	Experiment	ET	GB	MLP	RF	AB	K-NN
LMLS	A	**2.4**	24.4	6.0	3.7	2.66	4.76
	B	**2.4**	21.2	6.7	3.5	3.16	4.76
	C	**2.6**	21.2	4.5	3.0	2.87	4.76
	D	**2.3**	16.6	18.7	3.2	3.26	4.76
MAE	A	**243.1**	880.6	495.3	280.3	263.76	353.77
	B	**240.1**	868.9	620.1	300.8	317.70	353.77
	C	**249.5**	821.5	414.9	280.7	277.65	353.77
	D	**243.5**	768.2	855.9	283.5	336.57	353.77
MAPE	A	**29.2**	106.0	59.9	33.7	31.72	42.62
	B	**28.9**	104.6	74.9	36.2	38.24	42.62
	C	**30.0**	98.9	50.1	33.8	33.40	42.62
	D	**29.3**	92.5	103.2	34.1	40.52	42.62
MSE	A	**4.8**	48.8	12.0	7.4	5.32	9.53
	B	**4.9**	42.5	13.4	7.0	6.33	9.53
	C	**5.2**	42.5	9.0	6.1	5.75	9.53
	D	**4.6**	33.2	37.5	6.3	6.51	9.53
NMSE	A	508.4	535.7	979.6	370.6	572.03	**108.90**
	B	527.9	570.0	250.6	320.7	540.69	**108.90**
	C	561.2	569.8	867.9	407.3	537.47	**108.90**
	D	658.0	257.4	1883.6	428.9	502.79	**108.90**
SMAPE	A	**29.3**	106.3	59.9	33.7	31.75	42.67
	B	**28.9**	104.9	75.0	36.2	38.28	42.67
	C	**30.0**	99.1	50.1	33.8	33.44	42.67
	D	**29.3**	92.7	103.4	34.1	40.56	42.67

**Table 2 sensors-19-02740-t002:** Day model errors (multiplied by $10^{5}$) for extremely randomized trees(ET), gradient boosting (GB), multi-layer perceptron (MLP), random forest (RF), adaptive boosting (AB), and k-nearest neighbors (K-NN).

Error	Experiment	ET	GB	MLP	RF	AB	K-NN
LMLS	A	**2.9**	15.7	1153.2	3.3	3.56	5.81
	B	**3.2**	19.5	77.7	3.9	3.96	5.81
	C	**3.2**	29.5	16.3	3.9	4.15	5.81
	D	**3.1**	34.1	304.7	4.0	3.73	5.81
MAE	A	**232.0**	689.1	3079.4	269.9	355.44	363.79
	B	**255.1**	832.2	1515.3	318.7	332.95	363.79
	C	**280.8**	1102.7	727.6	321.4	483.44	363.79
	D	**278.4**	1136.0	1675.1	320.7	407.91	363.79
MAPE	A	**27.8**	82.8	372.6	32.4	42.75	43.75
	B	**30.6**	100.2	183.2	38.3	40.00	43.75
	C	**33.8**	132.8	87.8	38.6	58.27	43.75
	D	**33.5**	136.8	202.6	38.5	49.10	43.75
MSE	A	**5.9**	31.5	3457.9	6.7	7.12	11.62
	B	**6.3**	39.0	157.5	7.8	7.93	11.62
	C	**6.4**	59.1	32.6	7.9	8.30	11.62
	D	**6.2**	68.3	672.6	8.0	7.45	11.62
NMSE	A	799.6	783.7	18418.0	544.7	724.68	**98.50**
	B	425.2	347.0	7103.5	366.3	731.76	**98.50**
	C	459.0	193.2	1652.2	361.8	111.51	**98.50**
	D	434.8	147.3	13330.6	408.	733.02	**98.50**
SMAPE	A	**27.9**	83.0	349.5	32.4	42.80	43.82
	B	**30.7**	100.4	184.3	38.3	40.05	43.82
	C	**33.8**	133.1	88.0	38.7	58.32	43.82
	D	**33.5**	137.1	198.0	38.6	49.15	43.82

**Table 3 sensors-19-02740-t003:** Night model errors (multiplied by $10^{5}$) for extremely randomized trees (ET), gradient boosting (GB), multi-layer perceptron (MLP), random forest (RF), adaptive boosting (AB), and k-nearest neighbors (K-NN).

Error	Experiment	ET	GB	MLP	RF	AB	K-NN
LMLS	A	**0.05**	0.10	0.3	0.10	0.09	0.07
	B	**0.04**	0.10	3254.6	0.08	0.10	0.06
	C	**0.05**	0.10	0.10	0.07	0.09	0.06
	D	**0.04**	0.10	633.3	0.06	0.10	0.06
MAE	A	**38.6**	67.8	141.3	55.6	42.00	39.90
	B	**33.6**	57.6	14477.5	49.5	52.50	35.70
	C	**35.3**	65.1	110.6	48.1	42.00	37.80
	D	**33.3**	67.5	5595.9	45.9	52.50	37.80
MAPE	A	**4.7**	8.2	17.1	6.7	5.09	4.83
	B	**4.1**	7.0	1754.5	6.0	6.36	4.32
	C	**4.3**	7.9	13.4	5.8	5.09	4.58
	D	**4.0**	8.2	678.2	5.6	6.36	4.58
MSE	A	**0.11**	0.21	0.55	0.20	0.18	0.13
	B	**0.08**	0.21	6996.17	0.16	0.20	0.11
	C	**0.11**	0.20	0.20	0.14	0.18	0.12
	D	**0.08**	0.23	1291.10	0.13	0.20	0.12
NMSE	A	**4735.4**	8055.6	18847.7	7114.2	6805.56	822.22
	B	2407.6	**1427.0**	16796.8	5333.3	6388.89	1355.56
	C	**5349.2**	6283.4	22194.2	5732.9	6805.56	2355.56
	D	3829.9	7114.8	64166.1	**3324.2**	8055.56	2356.56
SMAPE	A	**4.7**	8.2	17.1	6.7	5.09	4.83
	B	**4.1**	7.0	1711.6	6.0	6.36	4.83
	C	**4.3**	7.9	13.4	5.8	5.09	4.83
	D	**4.0**	8.2	687.5	5.6	6.36	4.83

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
