# Peer review of "A Fault Detection System for a Geothermal Heat Exchanger Sensor Based on Intelligent Techniques"

_sensors, 2019, doi:10.3390/s19122740_

Round 1

Reviewer 1 Report

This paper presents a fault detection system for a geothermal heat exchanger system. The main goal was to define a strategy to identify a malfunctioning sensor and to establish a correct measured value in those scenarios. Here are some questions after reviewing the paper:

1. Line 18: Instead of saying in [2], the name of the authors should be mentioned

2. figure 4 is not necessary

3. can the algorithm find the best choice for the regressor if you change your case study?

4. How this strategy affects the overall error of the system?

5. More case studies should be presented 

]What do you mean by a default sensor? is this a sensor that is broken and should be changed or just have more uncertainty than the normal one?

Author Response

We are pleased to resubmit for the possible publication the new revised version of the

manuscript \A fault detection system for a geothermal heat exchanger sensor based on in-

telligent techniques" (sensors-513547) for the special issue Selected Papers from HAIS2018

in the journal Sensors.

We firstly wish to thank the issue and journal organization and the two reviewers for their

valuable contributions to this paper, which we have found very helpful. Their detailed

comments have enabled us to improve both the scienti c content of the paper and its

presentation. The revision of the manuscript is described in detail below.

Our revision has taken full account of the comments made by the editor and the two

reviewers. Our principal goal has been at all times to improve the scienti c contribution

of the paper, its readability and its overall presentation. We have also clari ed the novel

aspects of the work.

Our individual responses to each one of the reviewers' comments are set out as pdf file attached.

Reviewer 2 Report

The manuscript is well structured and the ideas properly presented.

The justification of the approach, as well as the analysis of the results is also well explained.

However, considering the results, some concerns appeared during the study of the paper.

First about the experiments setup. The authors explain that a 70/30 partition has been considered following classical approaches against overfitting. However, in multiple applications, a v-fold cross validation is applied in order to analyses possible data distributions dissimilarities. Are the test and training data sets statistically similar?

Second, and about the parameters for each considered technique, Could the authors provide references or similar works for which the considered values have been inferred?

Third, it would be interesting to know, from the authors point of view, the possibilities to consider a model data fusion level scheme to benefit from the different model responses.

Finally, fourth, what are the properties of the OC-SVM that could improve the sensor fault detection procedure in a future work?

Author Response

(The authors gave the same response as above.)

Round 2

Reviewer 1 Report

The authors have addressed all the comments and the paper can be published in the current format.